# Spatiotemporal dynamics and influencing factors of soil heterotrophic respiration in northeast China

**Dan Liu** [ORCID][1*○], **Cheng Long Yu**[1○], **Rui Feng**[2,3,4‡], **Shi Ping Yin**[1‡]

**1** Depatment of Meteorological, Heilongjiang Province Institute of Meteorological Sciences, Harbin, Heilongjiang Province, China, **2** Institute of Atmospheric Environment, China Meteorological Administration, Shenyang, Liaoning Province, China, **3** Shenyang Institute of Agricultural and Ecological Meteorology, Chinese Academy of Meteorological Sciences, Shenyang, Liaoning Province, China, **4** China Meteorological Administration, Key Laboratory of Agricultural Meteorological Disasters in Liaoning Province, Shenyang, Liaoning Province, China

○ These authors contributed equally to this work.
‡ RF and SPY also contributed equally to this work.
* nefuliudan@163.com

## Abstract

Soil heterotrophic respiration (Rh) represents a primary pathway of carbon release from soil. Using meteorological data, DEM, soil organic carbon density, and other data, we simulated the Rh in Northeast China from 2001 to 2020 using the GSMSR model. We then analyzed its spatialtemporal distribution pattern and examined its spatial-temporal aggregation, differentiation characteristics, and influencing factors at the national level, employing methods such as standard deviation ellipse (a statistical method that describes the spread and direction of data points in space), cold-hot spot analysis, and geographically weighted regression. The results showed that: (1) From 2001 to 2020, the annual mean Rh of the terrestrial ecosystem in Northeast China ranged from 24.22 kgC/ha/year to 25.02 kgC/ha/year, with a very significant increasing trend at the rate of 0.04 kgC/ha/year. The total amount of carbon release from soil heterotrophic respiration ranged from $4.76 \times 10^{11}$ to $5.02 \times 10^{11}$ kilograms per year (kg/ year), representing the annual carbon flux in the study region. And it had a significant increasing trend at the rate of $5.75 \times 10^{8}$ kg/year. (2) From the spatial differentiation and spatial clustering pattern, Rh was dominated by a northeast-southwest direction, its spatial distribution center was close to the northeast geographical center, and it had no obvious contraction or expansion trend on the whole. (3) In the northern and northeastern regions of the study area, vegetation cover directly influences local soil respiration rates. In most areas of the north, east, and south, per capita Gross Domestic Product directly affects soil respiration rates. It might provide a reference for the estimation of soil carbon loss and ecosystem carbon sink in this region.

**Data availability statement:** This study utilized several third party datasets, including the "MCD 12Q1" dataset, publicly available from the LAADS DAAC (Level-1 and Atmosphere Archive & Distribution System (https://ladsweb.modaps.eosdis.nasa.gov/search/order/1/MCD12Q1--61) after registration, the "China Surface Meteorological Observation Data" dataset, publicly available from the National Meteorological Information Center (China Meteorological Administration Meteorological Data Center) (https://data.cma.cn/data/cdcdetail/dataCode/A.0012.0001.S011.html) after registration, and the "SRTM 90m Digital Elevation Database" dataset, publicly available from the Consultative Group for International Agriculture Research Consortium for Spatial Information (https://bigdata.cgiar.org/srtm-90m-digital-elevation-data-base/). This study also used the third party "1:110m" dataset, publicly available from Natural Earth (https://www.naturalearthdata.com/downloads/110m-cultural-vectors/), and the "Harmonized World Soil Database v2.0" dataset, publicly available from the Food and Agriculture Organization of the United Nations (https://www.fao.org/soils-portal/data-hub/soil-maps-and-databases/harmonized-world-soil-database-v20/en/). The authors confirm that interested researchers would be able to access these data in the same manner as the authors. All other relevant data for this study are publicly available from the Dryad repository (http://doi.org/10.5061/dryad.9zw3r22s1).

**Funding:** This research program was generously supported by the Natural Science Foundation of Heilongjiang Province (Grant No.: LH2022D023), Innovation Development Project of China Meteorological Administration (Grant No.: CXFZ2023J059), Key Laboratory of Agrometeorological Disasters Joint Open Fund of Liaoning Provincial (Grant No.: 2023SYIAEKFMS27), Key Innovative Team of Agricultural Meteorology of the China Meteorological Administration (Grant No.: CMA2024ZD02), and Basic Research Fund of CAMS(2024Z001). Grant No.-LH2022D023 provided major funding support, access to large computational equipment, and meteorological data. Grant No.-CXFZ2023J059 provided partial funding support and access to soil data. Grant Nos. -2023SYIAEKFMS27 provided partial funding support and access to terrain, topography, and map data. Grant

## Introduction

Soil is a major carbon pool globally, and soil respiration (Rs) is the primary pathway for carbon release. Annual carbon emissions from Rs are estimated to be around 50–100 Pg [1], nearly 7–10 times the amount of carbon emitted from fossil fuels [2]. As Rs plays a critical role in the carbon cycle of terrestrial ecosystems, studying and assessing it is a key scientific issue in carbon cycle research. In recent decades, significant progress has been made in the regional observation and simulation of ecosystem carbon sequestration. However, data on Rs have not kept pace with these advancements. To our knowledge, no large-scale field measurement methods for Rs exist. Currently, Rs measurements are mainly conducted at the point scale or through multi-point networks, such as FLUXNET and ChinaFLUX, which are considered the most reliable methods for obtaining actual Rs values. Nevertheless, these approaches still carry several risks. For instance, some reports from eddy covariance (EC) tower observations [3] have indicated instances where Rs exceeded total ecosystem respiration, which contradicts biophysical theory. While Rs constitutes a major component of ecosystem respiration, it should not exceed the total respiration, which includes contributions from plant stems, leaves, and other aboveground parts [4]. This discrepancy may arise from underestimation of ecosystem respiration in flux tower measurements [5]. Laboratory calibration [6] can help bring the measurements closer to actual values, enabling the calculation of Rs at the point scale and providing essential data for the validation and parameterization of the Rs estimation models [7,8]. However, these data remain limited in their regional representativeness and ability to estimate Rs on a large spatial scale [9].

Estimation of Rs at the regional scale primarily relies on model simulations, which can be categorized into three main approaches: (1) Estimating total Rs for a specified region by calculating the average Rs for different land cover types and their respective areas [10]. This method is simple, but poor data can introduce substantial uncertainty in the Rs estimates. (2) Calculating Rh and analyzing its spatio-temporal variability based on process models. While this approach tends to produce relatively accurate estimates, the models are complex, require numerous parameters that are difficult to obtain, and their spatial representation remains uncertain. (3) There are several statistical models for estimating soil carbon flux based on environmental variables and measured soil carbon flux. One such model is the Geostatistical Model of Soil Respiration (GSMSR), which modifies Raich's global-scale statistical model [11]. Developed by Yu [8] using ChinaFLUX and other published datasets from 1995 to 2004, the GSMSR has been applied to estimate soil respiration rates across various terrestrial ecosystems, yielding satisfactory results [12].

Although Rs is commonly divided into Rh and Ra (Autotrophic Respiration), this simplification may not fully capture the long-term soil carbon turnover process. However, it aids in understanding soil respiration more easily [13]. Northeast China is located in a climate-sensitive region [14,15]. Its ecosystems, including forest, farmland, wetland and grassland are diverse. The region spans from the cold temperate zone to the warm temperate zone, and from the semi-arid to the semi-humid climate

Nos. -CMA2024ZD02 Key Innovative Team of Agricultural Meteorology of the China Meteorological Administration provided partial funding support. Grant Nos.-2024Z001 Basic Research Fund of CAMS provided partial funding support.

**Competing interests:** Ethical approval and consent to participate: Not applicable Consent to publish: Agree to publish **Competing interests:** The authors declare no competing interests.

zone. The sensitivity and vulnerability of ecosystems in Northeast China are among the highest in China and even in East Asia [16]. The complex climate and geographical environment, with its varied heat, humidity, and ecosystems, makes this region an ideal location for studying changes in soil heterotrophic respiration patterns in northern terrestrial ecosystems.

The objective of this study is to simulate the spatial-temporal dynamics of soil heterotrophic respiration in Northeast China from 2001 to 2020 using the GSMSR model and to analyze the influencing factors at a regional scale. Our hypotheses are: (1) Rh will exhibit significant spatial variation across different land use types, and (2) regional differences in vegetation coverage and economic development will influence the spatial patterns of Rh. By addressing these issues, this study aims to provide a reference for estimating soil carbon loss and ecosystem carbon sinks in the region, thereby enhancing our understanding of soil carbon fluxes in Northeast China.

## Materials and methods

### Site description

Northeast China comprises three provinces of Heilongjiang, Jilin, and Liaoning, and some regions, including cities of Chifeng, Tongliao, Hulun Buir, and Xing'an League, in the east of Inner Mongolia Autonomous Region. It stretches from 111.15°E to 135.09°E and from 37.95°N to 53.56°N. Its land area is 787,000 km$^2$ and its altitude ranges from 2 to 2,667 m. The area has a temperate monsoon climate, with cold and dry winters and hot and rainy summers. The annual average temperature over the past 30 years (1981–2010) has ranged from -4.08°C to 11.34°C. The annual rainfall ranges from 199.53 mm to 1170.60 mm (Fig 1). The average sunshine duration ranges from 5.26 h to 9.21 h[17]. Northeast China is surrounded by the Yalu River, Tumen River, Wusuli River, and Heilongjiang River to the east and north. In the interior, there are Daxing'anling Mountains, Xiaoxing'anling Mountains, Changbai Mountains, and Northeast China Plain (including the Songliao Plain, Liaohe Plain, and Three River Plain) which has formed a relatively independent geographical unit. The natural vegetation belongs to the Eurasian forest–grass plant subregion and the China–Japan forest plant subregion. Cold-temperate coniferous forests, temperate pine-broad-leaved mixed forests, warm-temperate deciduous broad-leaved forests, and vast grasslands are distributed in the area. The main crops are corn, rice, soybean and wheat.

The land cover types were obtained from the MCD12Q1 detaset provided by the LAADS DAAC (Level-1 and Atmosphere Archive & Distribution System https://lad-sweb.modaps.eosdis.nasa.gov/). MCD12Q1 includes various classification schemes that describe land cover properties based on a year's worth of Terra and Aqua satellite data with 500m pixel resolution. In this study, we uesed the IGBP classification system, which consists of 11 natural vegetation classes, 3 developed and mosaicked land classes, and 3 non-vegetated land types. these IGBP calsses were further combined into 7 categories for our analysis (Table 1).

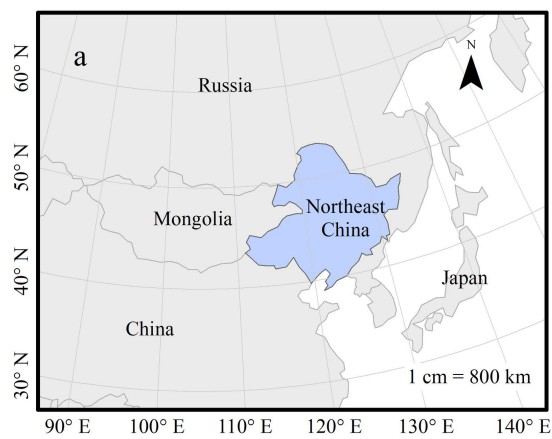

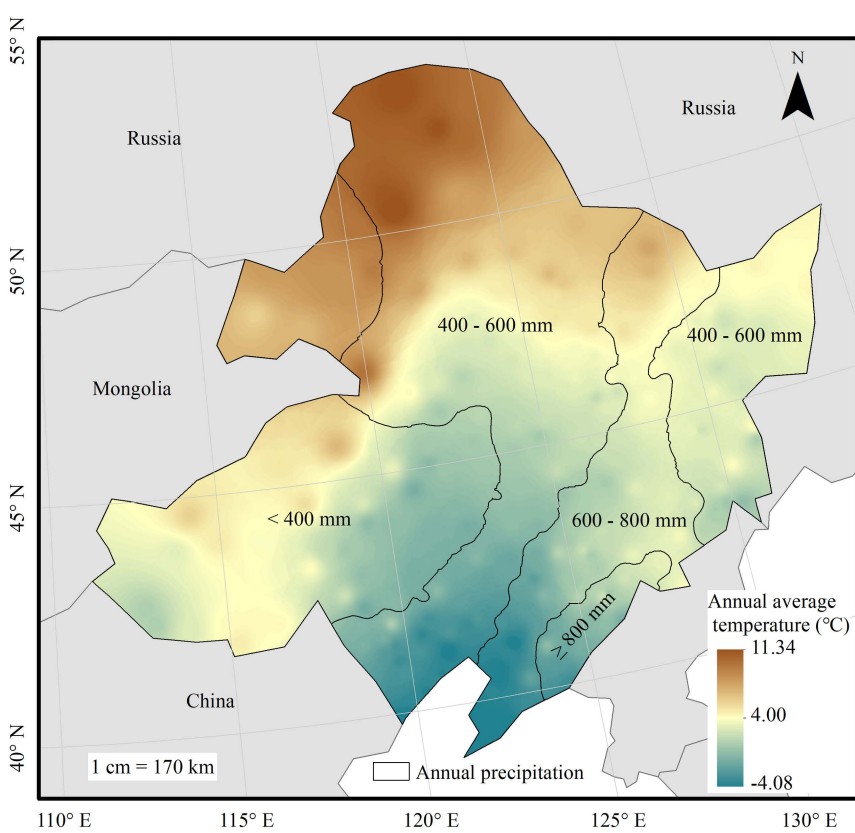

**Fig 1. Overview of the study areaLand cover type.**

**Table 1. The corresponding relationship between land cover types in this study and the classification system of IGBP.**

| ID | Land use | Class of IGBP |
|---|---|---|
| 1 | Farmland | croplands, cropland/natural vegetation mosaics |
| 2 | Forest | evergreen needleleaf forests, evergreen broadleaf forests, decidnous needleleaf forests, mixed forests, closed shrublands, open shrublands, woody savannas, savannas |
| 3 | Grassland | grasslands |
| 4 | Wetland | Permanent wetlands |
| 5 | Water | Water bodies |
| 6 | Artificial surface | Urban and built-up lands |
| 7 | bareland | Permanent snow and ice, barren, unclssified |

## Meteorological observation data

The temperature and precipitation data are both from the monthly climate data set of China Meteorological Administration (http://data.cma.cn/). We used ANUSPLIN [17,18] software package to rasterize the meteorological data. Three statistical parameters were selected to evaluate the consistency between the simulated values and the real values of the before and after calibration [19].

Root mean square error, RMSE

$$RMSE = \sqrt{\frac{\sum_{i=1}^{n}(S_i - O_i)^2}{n}}$$

(1)

Model simulation efficiency, E

$$E = 1 - \frac{\sum_{i=1}^{n}(O_i - S_i)^2}{\sum_{i=1}^{n}(O_i - \overline{O})^2}$$

(2)

Consistency index, d

$$d = 1 - \frac{\sum_{i=1}^{n}(O_i - S_i)^2}{\sum_{i=1}^{n}(\left|S_i - \overline{O}\right| + \left|O_i - \overline{O}\right|)^2}$$

(3)

Where $n$ is number of samples; $S_i$ and $O_i$ is simulated and measured value respectively; $\overline{O}$ is measured average value. When RMSE is closer to 0, the simulation values are closer to the real values and the simulation effect is better. $E \in (-\infty, 1)$, the closer the E value is to 1, the closer the simulated values are to the true values; $E \in (-\infty, 0)$ represents that the average values of the observed values are higher than those of simulated prediction. $d \in (0, 1)$, the closer d is to 1, the better the consistency between the simulated and the real values are.

The temperature observation data of 926 meteorological observation stations which did not participate in the simulation from 2009 to 2020 were used to verify the temperature simulation effect, and the two sets of data had a very significant linear relationship (RSME = 1.5505, E = 0.9885, d = 0.9899, Fig 2). It can be seen that the temperature data fitted by

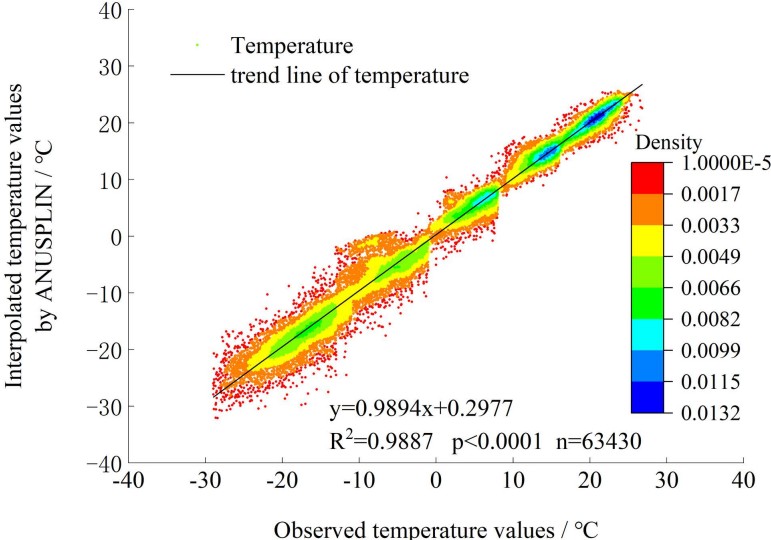

**Fig 2. Comparison of interpolated temperature values and observed temperatureDigital elevation model (DEM) and map data.**

ANUSPIN has a good applicability in Northeast China. Due to the lack of effective precipitation test data in this study, the precipitation fitting results were not verified.

The DEM dataset was available at the website of Consultative Group for International Agriculture Research Consortium for Spatial Information (http://srtm.csi. cgiar.org/). This version of the Shuttle Radar Topography Mission (SRTM) digital elevation data has been processed to fill data voids, and to facilitate its ease of use [20]. The website provides data at spatial resolution approximately 30m and 90m. We downloaded the 90m resolution data covering the study area and resampled to 500m for consistency with other datasets used in this study, such as the MCD12Q1 land cover dataset with 500m resolution, and the soil organic carbon density data with 1km resolution. This resampling ensured uniformity in spatial resolution across all datasets, which is important for accurate spatial analysis and model integration.

The map data is sourced from vector data provided by National Catalogue Servic for Geographic Information.

## Soil organic carbon density

The data were collected from the world soil database (https://www.fao.org/soils-portal/en/), which was constructed by FAO and IIASA. In this study, with a spatial resolution of 1 km, the soil organic matter content in 0–30 cm depth was used to participate in the GSMSR model.

To prepare for the calculation of soil monthly respiration by using GSMSR, soil organic carbon content was converted to soil organic carbon density by using the following formula [21]:

$$D_s = \sum_{i=1}^{n} T_i \times BD_i \times SOC_i \times \frac{(1 - C_i)}{100}$$

(4)

Where Ds is the organic carbon content (kg/m$^2$); $T_i$ is the thickness of the i-layer soil (cm); BDi is the bulk density of the i-layer soil (g/cm$^3$); SOCi is the organic carbon content (g/kg) and Ci is the proportion of gravel size larger than 2mm in i-layer soil(%).

## GSMSR

Although the GSMSR model requires substantial spatial data for precise modeling, is computationally complex, and demands high-quality data, it is based on geostatistical methods that can capture the spatial heterogeneity of soil respiration. It is capable of adapting to complex geographical and climatic conditions, and is well-suited for long-term time-series data analysis. Compared to process models such as BIOME-BGC [22] and DAYCENT [23], which involve more complex parameterization, GSMSR offers a simpler parameter framework. Moreover, it more effectively captures the spatial heterogeneity and intricate ecological variations of soil respiration compared to simplified models like CASA [24] and Q10 [25]. Unlike machine learning and statistical models [26], it does not rely on large training datasets and offers superior interpretability." GSMSR [8] was developed by modifying a global scale statistical model, which is driven by monthly air temperature, monthly precipitation, and soil organic carbon (SOC) density, can capture 64% of the spatiotemporal variability of soil respiration.

$$R_{s,monthly} = (R_{Ds=0} + MDs)e^{\ln \alpha e^{\beta T/10}} (P + P_0)/(P + K)$$

(5)

Where $R_{s,monthly}$ is monthly mean $R_s$ rate (gC·m$^{-2}$·d$^{-1}$), $D_S$ is SOC density at a soil depth of 20 cm, $R_{Ds=0}$ is $R_{s,monthly}$ rate when the SOC density is zero, $P_0$ is capability of water retention in soil, $T$ is monthly air temperature, $P$ is monthly precipitation, $M$——parameter. The values of parameters are $R_{Ds=0} = 0.588$, M = 0.118, $\alpha = 1.83$, $\beta = -0.006$, $P_0 = 2.972$, and $K = 5.657$ Chuai [27]. In subsequent analyses, the unit of $R_{s,monthly}$ was converted to kgC/ha/year.

Researches have shown that the static chamber/GC method and static/dynamic chamber/IRGA methodd provides reliable localized soil respiration data across different soil types and climatic conditions [28,29]. To compare with the results simulated by the GSMSR model, this study summarizes 28 articles published between 2003 and 2022 on soil respiration

measurements in Northeast China. Only studies employing the static chamber/GC method and static/dynamic chamber/IRGA method were selected. Monthly average data for 537 soil respiration rates and 163 soil heterotrophic respiration rates (from May to September) were collected to calculate the mean soil respiration rate. The average soil respiration rate was then calculated for both the ecosystem and the study site. Because different studies repoted soil respiration in various units, all values were converted to a uniform unit of (kgC/ha/year). These data included observations from four ecosystems: forest, farmland, grassland and wetland. Due to the lack of geographical coordinates for most of the observation soil sites, we could only define the approximate observation range based on the descriptions in the leterature. We then compared and analyzed the soil respiration data simulated by the GSMSR model with the measured data within the observation range. The results, shown in Table 2, indicate that the range of simulated value (11.88 kgC/ha/year) is lower than the measured value (14.90 kgC/ha/year), but the average value are quite similar::simulated value is 26.61 kgC/ha/year, and the observed value is 26.68 kgC/ha/year. Therefore, we conclude that the soil respiration model can be used to simulate the soil respiration rate in Northeast China. The differences between the simulated and measured values are relatively small across different ecosystems, indicating that the GSMSR model can predict soil respiration rates with considerable accuracy. Notably, in wetland and certain forest ecosystems, the simulated results closely match the measured values, suggesting that the model is applicable to various land use types.

The GSMSR model was driven by environmental variables with the two following constraints. The soil respiration rate was constant when the soil organic carbon density was larger than or equal to 15.09 kg/m² at 20 cm soil layer. The soil respiration rate was zero when the environmental temperature was less than -15.96°C, and it was the maximum when the environmental temperature was more than 30.96°C [1,8].

The proportion of the average soil heterotrophic respiration rate to the total respiration rate in 163 months was calculated firstly (Fig 3-a). The results showed that the proportion in March, November and December was between 82.47% and 93.72%, which was obviously higher than that in April to October (59.86% -73.48%). The soil respiration rate and heproportion from January to December was calculated according to the fitted quadratic curve equation and GSMSR respectively (Fig 3-b). Then monthly average soil heterotrophic respiration rate in northeast China was calculated and added throughout the year.

### Temporal variations of soil heterotrophic respiration during 2001–2020

From 2001 to 2020, the annual mean soil heterotrophic respiration rate of the whole terrestrial ecosystem in the study area was between 24.22–25.02 kgC/ha/year, with an average of 25.01±0.43 kgC/ha/year. There was a significant increasing trend by 0.04 kgC/ha/year.

To verify the significance of the differences in annual mean soil respiration and carbon release among different land use types, we conducted a non-parametric independent sample test. The results showed that there were significant

**Table 2. Comparison of soil respiration simulated and measured values.**

| Systems | Site | Simulated values of soil respiration rate (kgC/ha/year) | Measured values of soil respiration rate from literatures (kgC/ha/year) |
|---|---|---|---|
| Forest | Daxing'anling Mountains | 21.15 | 23.31 [30–34] |
| | Maoershan experimental forest farm | 30.05 | 28.17 [35–38] |
| | Xiaoxing'anling Mountains | 29.48 | 29.87 [39–42] |
| | Changbai Mountains | 30.04 | 31.87 [43–47] |
| Farmland | Songnen Plain | 28.74 | 26.49 [48–53] |
| Wetland | Daxing'anling Mountains | 18.17 | 16.97 [54] |
| Grassland | Hulun Buir grassland | 28.62 | 30.07 [55] |

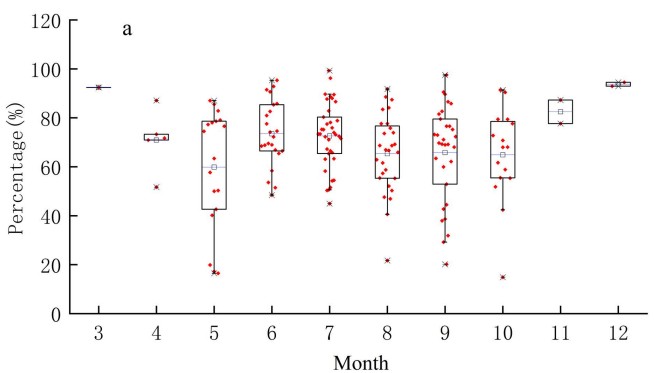
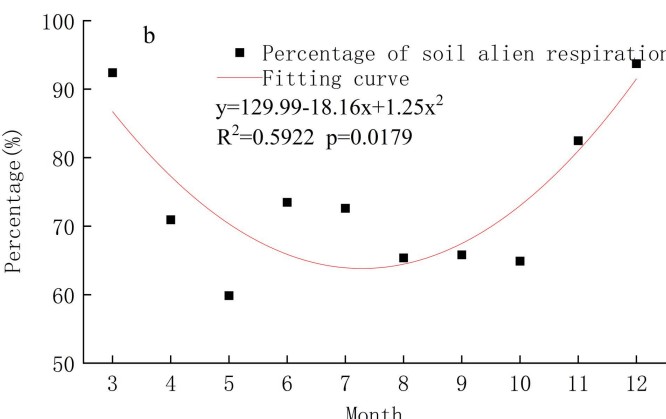

**Fig 3. The proportion of soil heterotrophic respiration rate to total soil respiration rate (a) and proportion average (b) Results.**

differences in the annual mean soil respiration and carbon release among the land use types (P < 0.001). The order from high to low was farmland (32.88–34.99 kgC/ha/year, average 33.95 kgC/ha/year), wetland (30.92–34.69 kgC/ha/year, average 32.83 kgC/ha/year), grassland (27.70–29.52 kgC/ha/year, average 28.57 kgC/ha/year), forest (27.92–29.22 kgC/ha/year, average 28.53 kgC/ha/year) and bare land (22.47–24.97 kgC/ha/year, average 23.88 kgC/ha/year). High soil organic carbon input (crop residues, fertilization), frequent soil disturbance (tillage) which increases aeration and microbial activity, and stable soil moisture conditions conducive to microbial growth; the combined effects of these factors result in the highest soil heterotrophic respiration in farmland among various land use types.

The total amount of carbon release by soil heterotrophic respiration ranged from $4.76 \times 10^{11}$–$5.02 \times 10^{11}$ kg/year, with an average of $4.88 \times 10^{11} \pm 7.10 \times 10^{9}$ kg/year. There was a significant increasing trend by $5.75 \times 10^{8}$ kg/year. The annual mean carbon release was grassland ($1.71 \times 10^{11}$ kg), forest ($1.63 \times 10^{11}$ kg), farmland ($1.53 \times 10^{11}$ kg), bare land ($7.71 \times 10^{8}$ kg) and wetland ($6.44 \times 10^{8}$ kg) in order from large to small. Among them, the carbon release of forest, wetland and farmland showed a very significant increase trend, and that of grassland showed a very significant decrease trend. The carbon release of bare land decreased sharply before 2003, and then relatively stable (Fig 4).

**Spatial differentiation characteristics**

The standard deviation ellipse method was used to analyze the spatial differentiation pattern of Rh (Fig 5). The overall direction of Rh was dominated by the northeast-southwest direction and the azimuth change range was 46.48 ° - 50.29 °. The spatial distribution center was close to the geographic center of Northeast China (123.21 °E, 46.07 ° N) with a distance of 13.41–24.96km, and the spatial distribution range has no obvious change.

The azimuth angle can be used to identify the spatial aggregation characteristics of Rh. It could be seen that the forest Rh was dominated by the northwest-southeast direction and the average azimuth angle was 148.97°with a significant increasing trend, which indicated that the dominant direction of forest Rh shifted in the south-north direction (Fig 5-a). The dominant direction of other ecosystems was in the northeast-southwest direction, of which the azimuth angle of farmland Rh obviously shifted in the east-west direction. The distance between the spatial distribution center of bare land and the geographic center was the largest with an average of 559.14km and that of wetland was the smallest with an average of 178.87km. In addition, the distribution centers of grassland (Fig 5-b) and farmland (Fig 5-e) tended to move westward, that of forest (Fig 5-a) tended to move southwest, and that of wetland (Fig 5-c) tended to move northeast. The forest had the largest spatial distribution range, covering 42.16% of the total area in Northeast China, and showed a significant increasing trend. In concrast, both wetland and bare land exhibitde clear shrinking trend, whele farmland and grassland showed

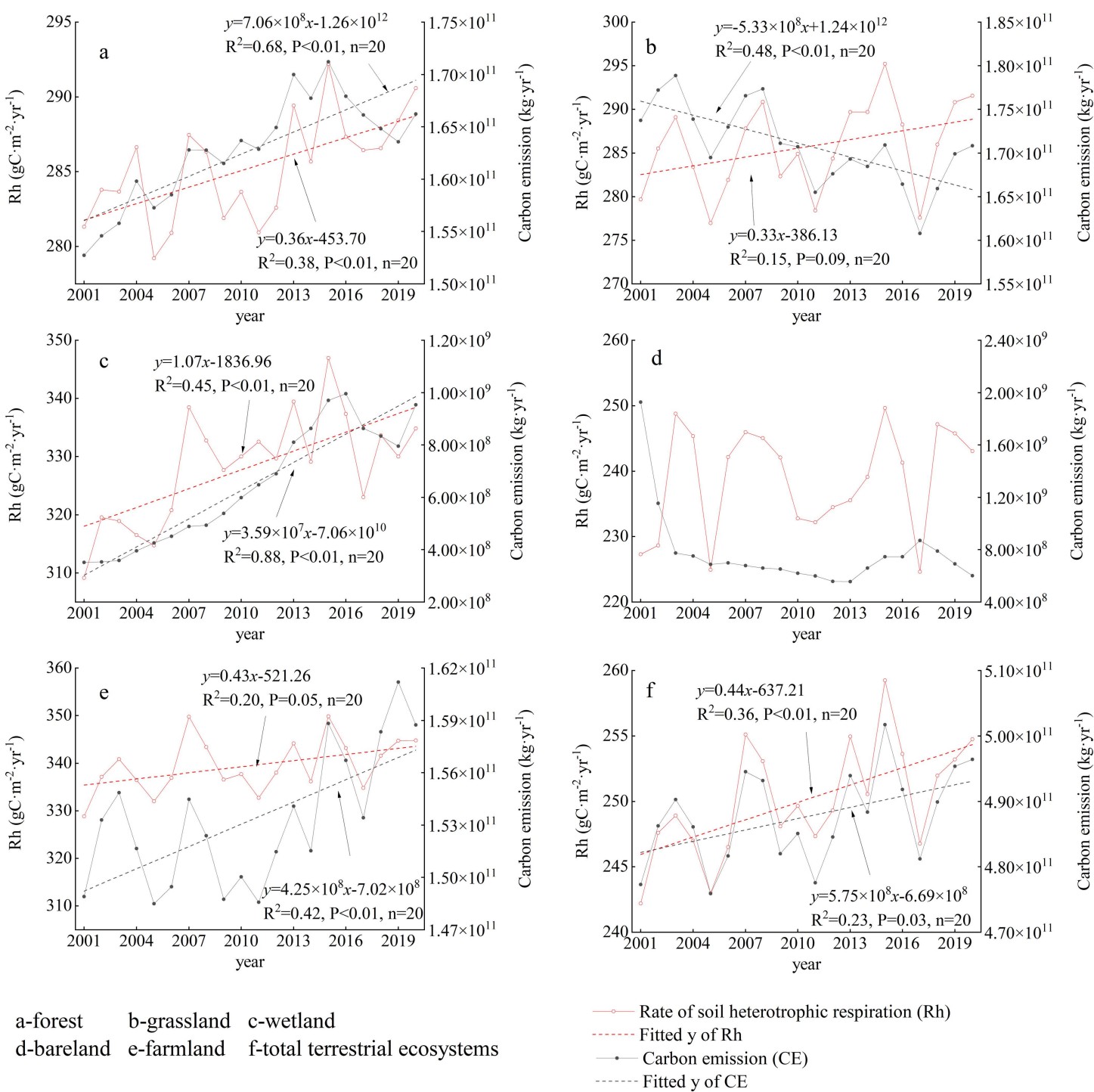

**Fig 4. Soil respiration rate and carbon release in different land use types.** The carbon release in the figure is generated by soil respiration, and calculated by multiplying area of each land use types by the soil heterotrophic respiration rate.

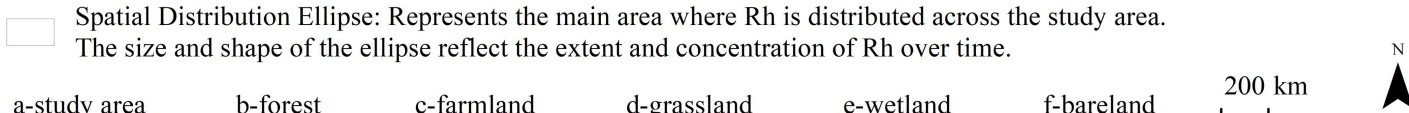

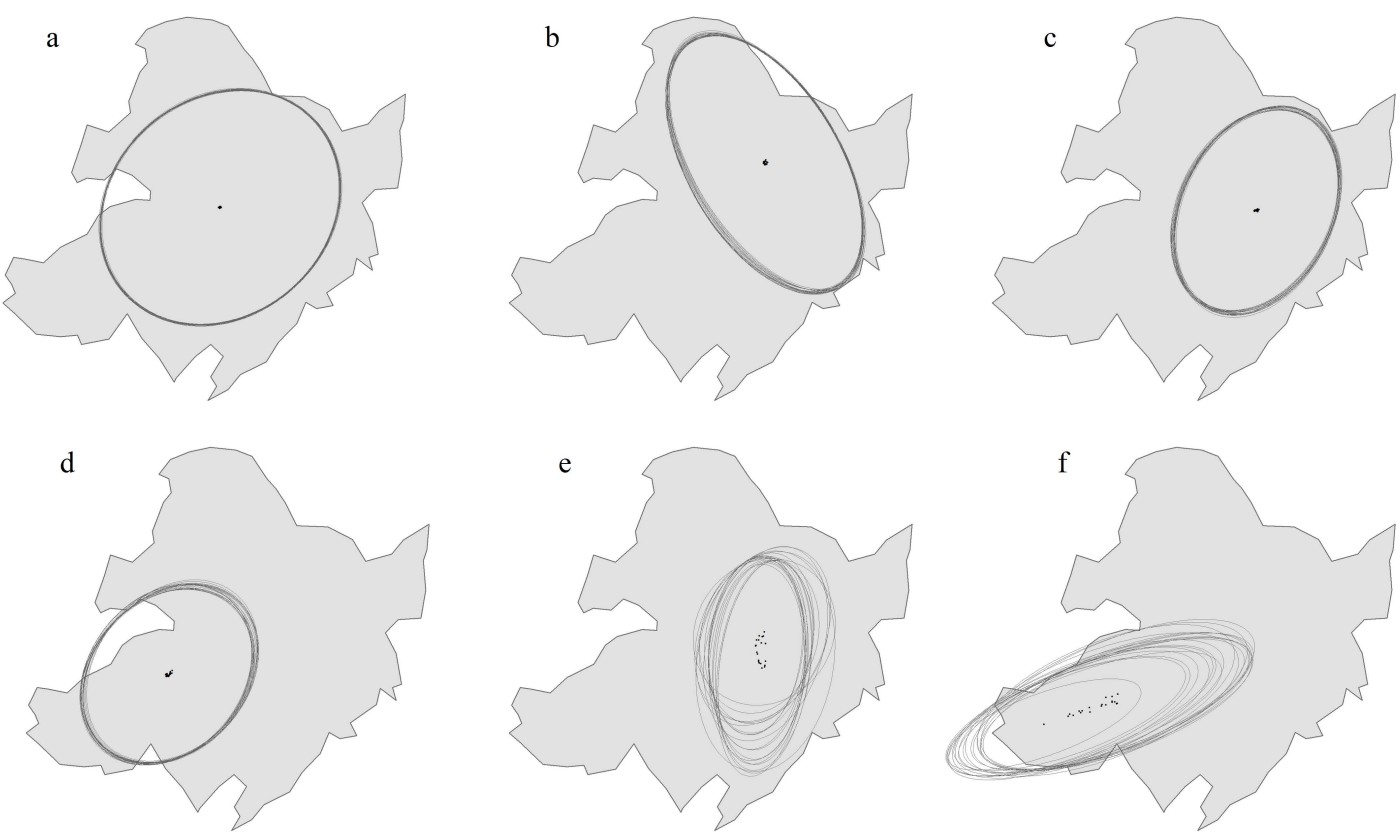

- Center of the Ellipse: Represents the central location of the distribution of Rh in the two-dimensional space of the study area.

Spatial Distribution Ellipse: Represents the main area where Rh is distributed across the study area.
The size and shape of the ellipse reflect the extent and concentration of Rh over time.

a-study area        b-forest        c-farmland        d-grassland        e-wetland        f-bareland

200 km        N

**Fig 5. The specific ellipse of soil heterotrophic respiration rate in Northeast China.** The study area was divided into 50×50km, and the geographical center point of each segmented area was taken as the sample points. Such 56,469 sample points were defined to obtain the annual mean soil respiration rate from 2001 to 2020. Using the standard deviational ellipse (SDE) method, the spatial distribution ellipse of soil heterotrophic respiration rate (Rh) in Northeast China from 2001 to 2020 was calculated with Rh as the weight. The spatial range of the ellipse represented the main area of the spatial distribution of Rh, the center of the ellipse represented the central position of the distribution of Rh in the two-dimensional space, the direction of the long axis of the ellipse represented the main trend direction of the distribution of Rh, and the size of the long axis of the ellipse represented the dispersion degree of Rh in the main trend direction.

no significant changes. The axial ratio, defined as the ratio of the long axis to the short axis of the polarization ellipse, was highest for Rh in bare land, an average of 3.00, and showed an upward trend. Conversely, the axial ratios of Rh in other ecosystems demonstrated a downward trend.

## Spatial aggregation characteristics

As the direct unit of urban and rural land management in China, the impact of soil respiration of terrestrial ecosystems on atmospheric greenhouse gas content cannot be ignored at country level. It is of great significance to study the scale

and variation of soil carbon release at county level for the construction of land spatial organization system based on low-carbon orientation and the optimization of land spatial pattern.

By analyzing the cold and hot spots' spatial distribution characteristics of Rh every 5 years (Fig 6) and the mean heterotrophic respiration rate of soil (Table 3), the aggregation characteristics of these spots of annual mean Rh at county level in Northeast China were relatively stable. Overall, the Rh hot spots were located in the central region of Northeast China, with an average of 33.31±2.84 kgC/ha/yr from 2001 to 2020. Among them, the average was the hightest between 2016 and 2020, with 34.06±2.89 kgC/ha/year, The terrain of this area was dominated by plains and mainly distributed with farmland ecosystem. This region has abundant soil organic matter, coupled with frequent soil disturbance and suitable soil moisture and temperature conditions, making it Rh hot spots. The Rh cold spots were located in the southwest and south of Northeast China, with the average of 26.80±4.23 kgC/ha/year from 2001 to 2020, and the highest average of 27.83±4.61 kgC/ha/year from 2016 to 2020. The terrain in this area was mainly plain and plateau, and mainly distributed with grassland ecosystem and farmland ecosystem. The relatively low soil organic matter content is likely the main reason.

In spatio-temporal variation, the cold spots had little change and the hot spots in the central region were relatively stable from 2001 to 2020, and the hot spots gathered in Sanjiang Plain from 2016 to 2020. So the spatio-temporal distribution

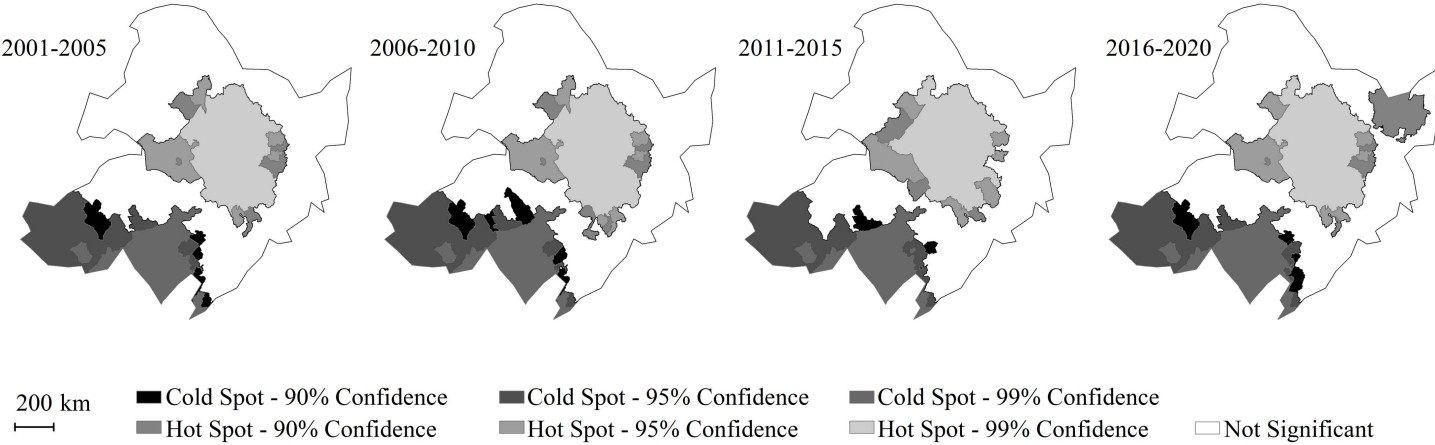

**Fig 6. Hot spot distribution of Rh at county level in Northeast China from 2001 to 2020.** The Getis-Ord Gi * index was used to analyze the local spatial dependence and heterogeneity of average Rh at country level in Northeast China. The index indicated a high-value cluster and was defined as hot spot region when its value was larger than zero. It indicated a low-value cluster and was defined as cold spot region when its value was smaller than zero.

**Table 3. Mean value of soil heterotrophic respiration rate in hot spots (or cold spots) distribution area from 2001 to 2020. Unite: kgC/ha/year.**

| Confidence | 2001-2005 | 2006-2010 | 2011-2015 | 2016-2020 |
|---|---|---|---|---|
| Cold Spot - 90% Confidence | 28.00±4.01 | 28.63±7.22 | 24.17±7.39 | 30.34±5.00 |
| Cold Spot - 95% Confidence | 27.00±3.43 | 26.92±2.45 | 28.18±3.92 | 27.57±5.53 |
| Cold Spot - 99% Confidence | 25.01±2.81 | 25.10±2.81 | 25.10±2.86 | 25.56±3.29 |
| Hot Spot - 90% Confidence | 30.78±2.68 | 32.04±3.71 | 29.59±0.93 | 33.37±2.60 |
| Hot Spot - 95% Confidence | 32.42±3.04 | 32.64±2.98 | 33.32±3.66 | 32.69±3.16 |
| Hot Spot - 99% Confidence | 35.24±2.79 | 35.78±2.83 | 35.76±2.78 | 36.09±2.89 |
| Not Significant | 30.42±3.47 | 30.85±3.19 | 30.83±3.18 | 30.63±2.92 |

pattern of the annual mean Rh in Northeast China was more in the center and less in the south, and the spatial aggregation was stable and rising.

It could be seen from the analysis of the spatial aggregation characteristics of Rh (Fig 7) that the annual Rh in Northeast China has the characteristics of high value and low value aggregation, high-low and low-high regional scattered distribution. Comparing the distribution characteristics of cold and hot spots of Rh (Fig 6), in addition to some hollow phenomenon of high values, other high values are basically consistent with hot spots, low values and cold spots. The "hollowing" of high values is mainly located in the southwest of Heilongjiang Province and the junction with Jilin Province.

## Analysis of influencing factors

The GSMSR model has already incorporated temperature and precipitation as key drivers in simulating soil heterotrophic respiration (Rh), meaning the impact of these meteorological factors is implicitly reflected in the model's output. Therefore, we chose to focus on factors that may independently influence Rh beyond the model's driving variables. Specifically, we believe that vegetation cover and the level of economic and social development could have significant effects on Rh, which may not have been fully explored in previous studies. By analyzing these factors, we aim to provide a more comprehensive perspective on the various influences on Rh. We applied the GWR method to analyze the spatial heterogeneity of the impact of natural and anthropogenic factors on Rh, considering the spatial relationships at the county level.

Based on the analysis of the relationship between Rh rate and vegetaion coverage in Fig 8-a, 8-b and Table 4, it can be seen that there was a significant (extremely significat) correlation between Rh and vagetation coverage in the north and northeast of the study area, indicating that vagetation coverage directly affects Rh in this area. At the same time,there was a negative correlation between the two in this area, which may be due to the decarease of soil aeration caused by high vagetation coverage, thus reducing the rate of Rh [56,57]. However, there was no significant correlation between Rh and vagetaion coverage in other regions, indicating that vegetation coverage did not play a dominant role in this region. There were many other important factors affecting Rh, which might include availability of soil organic matter, temperature, humidity, etc.

Based on the analysis in Fig 8-c, 8-d and Table 5, it can be seen that there was a significant (or extremely significant) positive correlation between Rh and the per catita GDP in most of the northern, eastern and southern regions of the study

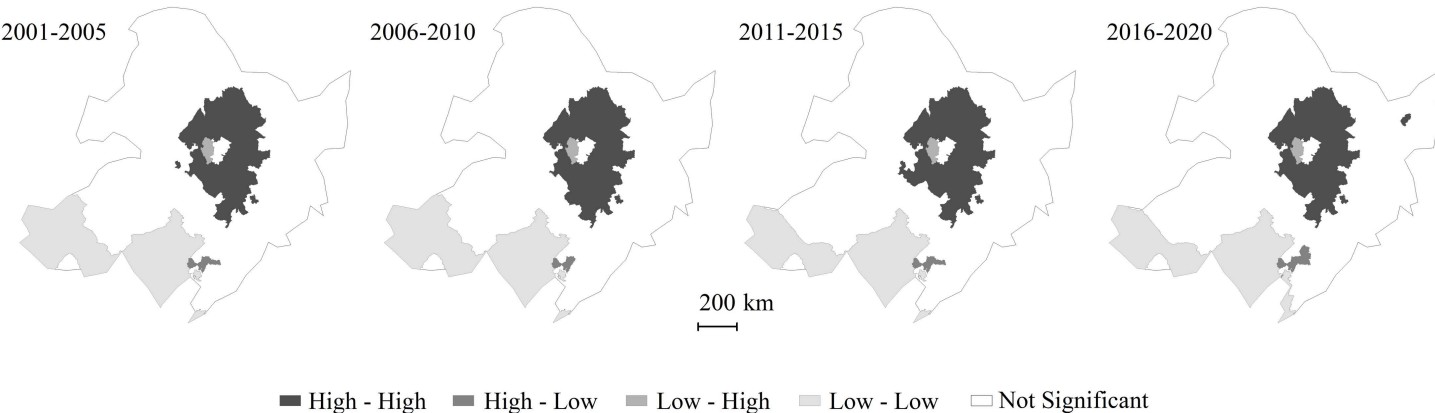

**Fig 7. Clustering characteristics of Rh at county level in Northeast China from 2001 to 2020.** The local Moran's I index was used to analyze the spatial clustering characteristics of annual mean Rh at country level in Northeast China. High-high indicated the regions of high values, low-low indicated the regions of low values, high-low indicated the regions of high values surrounded by low values, and low-high indicates the regions of low values surrounded by high values. The confidence of the above statistical tests was 95%.

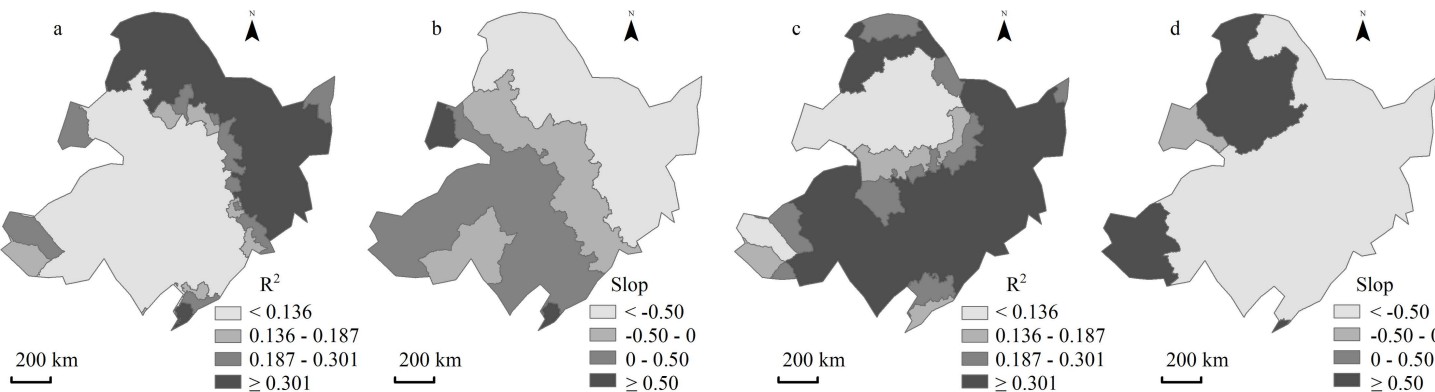

**Fig 8. Spatial distribution of determination coefficient (a) and slope (b) of linear regression between vegetation coverage and soil heterotrophic respiration, determination coefficient (c) and slope (d) of linear regression between per capita GDP and soil respiration rate based on GWR model from 2001 to 2020.** The statistical period is from 2001 to 2020, with a degree of freedom of 19. According to the significance test table of the correlation coefficient, $R^2 \geq 0.301$ is the most significant correlation, and $0.187 \leq R^2 < 0.301$ is the most significant correlation.

**Table 4. Mean value of regional soil heterotrophic respiration rate under different determination coefficients from 2001 to 2020. Unit: kgC/ha/year.**

| p-value | $R^2$ | NDVI_Rh | GDP_Rh |
|---|---|---|---|
| 0.10 | <0.136 | 24.02 | 23.96 |
| 0.05 | 0.1361-0.187 | 23.22 | 25.56 |
| 0.01 | 0.1871-0.301 | 25.98 | 23.63 |
| <0.01 | >0.301 | 23.33 | 23.90 |

*The calculation method of the average value of regional soil respiration rate is as follows: establish a linear regression equation with the Rh in the county as the dependent variable and the vegetation coverage in the corresponding region as the independent variable, calculate the determination coefficient $R^2$, divide $R^2$ according to the level of the correlation coefficient significance test table, and count the average value of Rh in different levels of $R^2$ regions.

**Table 5. Mean value of regional soil heterotrophic respiration rate under different determination coefficients from 2001 to 2020. Unit: kgC/ha/year.**

| slope coefficient | NDVI_Rh | GDP_Rh |
|---|---|---|
| <=-0.5 | 25.00 | 24.05 |
| <=0 | 25.48 | 23.99 |
| >0 | 21.98 | |
| >0.5 | 20.88 | |

*The calculation method of the average value of regional Rh is as follows: establish a linear regression equation with the soil respiration rate in the county as the dependent variable and the vegetation coverage in the corresponding area as the independent variable to obtain the slope coefficient, divide the slope coefficient into grades, and count the average value of Rh in different grade slope coefficient areas.

area, indicating that the local economic activities had a significant impact on the Rh. For example, the increase in crop cultivation area and the use of organic fertilizers have boosted local GDP (https://data.stats.gov.cn/), while also providing a continuous carbon source for microbial communities, promoting an increase in Rh [57,58].

## Discussion

### SOC changes were complex

This study inferred Rs on the assumption that SOC was constant. According to the GSMSR model, the change of Rs mainly depended on the change of meteorological factors, which was reasonable in a certain range, because soil organic matter undergone the transformation of mineralization and humification intermittently under the action of microorganisms. In the long term, the total SOC changed little or unchanged [59]. However, the SOC mineralized substrate fluctuated [60], and the increase of ecosystem GPP would lead to the increase of C absorption and litter. The change could increase the available carbon for soil microbial metabolism, the SOC mineralized substrate [61]. Meanwhile low organic matter input might reduce the quality and quantity of SOM, but might improve the temperature sensitivity of SOC mineralization [62], Unstable organic matter addition also increased the temperature sensitivity of SOC decomposition [63]. The increase in temperature might drive the enhancement of SOC mineralization and increase Rh. The Bond-Lamberty's [1] study showed that when the global average temperature increased by 0.7 °C in 25 years, Rh increases by 12%. Changes in nutrient cycling process and microbial community composition also had an impact on SOC mineralized substrate yield [64].

While assuming constant SOC simplifies the dynamics and complexity of the soil respiration process, it overlooks the fluctuations in the SOC mineralized substrate and the changes in temperature sensitivity. This assumption may introduce biases in the model's long-term predictions, responses to climate change, and estimates of the impacts of microbial community composition and nutrient cycling, thereby affecting the accurate prediction of soil respiration (Rh) rates.Therefore, strict and real-time SOC change measurement was necessary because there was no global real-time SOC map list at present. This required collaborative efforts to improve measurement methods, test verification, and reporting mechanism. However, as more systematic ecological observation station networks are established and additional data on organic matter reserves and flux observations become availible, the accuracy and regional representativeness of Rh measurement and simulation results will improve.

### Ecosystem carbon balance assessment

During the analysis of the net primary productivity (NPP) of terrestrial ecosystems in the Northeast region of China from 2001 to 2020, where NPP is defined as the amount of carbon assimilated by plants through photosynthesis minus the carbon released through plant respiration, we utilized the MOD17A3HGF dataset. The findings revealed that the average annual NPP value for the region ranged between 267.48 and 331.59. This value was compared with the average soil heterotrophic respiration rate, which was between 242.19 and 250.24, uncovering a notable ecological phenomenon: the NPP of the study area overall exceeded the soil heterotrophic respiration rate, indicating that the region was in a state of carbon sequestration. Further examination of the carbon absorption or release across different ecosystems revealed that forests, grasslands, and croplands were all in a state of carbon absorption, with forests making the most significant contribution, absorbing an average of $1.41 \times 10^{11}$ kg/year annually. In contrast, wetlands and bare lands were in a state of carbon release, with bare lands having the largest annual release, amounting to $5.29 \times 10^8$ kg/year.

### Changing soil carbon flux by optimizing land use

Rh of terrestrial ecosystem showed spatial-temporal inconsistency and significant differences under different ecosystem types, climate conditions, and human management pratices. It is well known that the quality and quantity of soil organic matter directly affect Rh [65–67]. These differences were caused by variations in species composition, large-scale communities changes, or the difference in ecosystem types [68], and were largely reflected in vegetation coverage.

Neff & Hooper's study [69] showed that the impact of vegetation types' change on quality and quantity of soil organic matter was more direct than that of climate change. The findings of this study supported this conclusion. There was a

strong positive correlation between vegetation coverage and Rh. Among them, the regions with the largest regression coefficient was the main forest regions in the Daxing'an Mountains and Xiaoxing'an Mountains, which indicated that the change of the northern forest ecosystems (cold temperate coniferous forest region) had a greater impact on Rh than that of the southern forest (temperate coniferous-broadleaf mixed forest region) and other ecosystems. In addition, the impact of vegetation coverage on Rh in the northern farmland ecosystem (Sanjiang Plain) was greater than that in the southern farmland ecosystem (Liaohe Plain). The reason might be that the soil organic carbon content mainly depended on the surface vegetation and land use type, and was closely related to the plant litter entering the soil and the microbial species in the soil [70]. If the vegetation coverage was high, this meant that more litter had entered the soil carbon pool so as to provide a more sufficient and mineralized substrate for Rh [61]. But this was not the only influencing factor, and temperature, soil moisture, soil microbial species, and etc. would also have an important impact.

The human disturbance to the farmland ecosystem was much greater than that of other natural ecosystems. For instance, intensive fertilization increases soil nutrient levels, stimulating microbial activity and consequently enhancing Rh. The use of pesticides can alter the structure of soil microbial communities, affecting the decomposition of organic matter and, in turn, influencing the rate of Rh. Irrigation practices regulate soil moisture, directly impacting microbial activity and Rh, particularly in areas with uneven precipitation. Additionally, straw returning increases the input of organic matter into the soil, providing more carbon sources for microbial metabolism, thus promoting an increase in Rh. Therefore, the sensitivity of Rh to changes in vegetation coverage in northern and southern farmland ecosystems, especially under different human management practices, requires further investigation to better understand the specific impacts of these management measures on Rh.

Land use optimization can help mitigate Rh variability by promoting more sustainable agricultural and forestry practices that enhance soil organic matter content and microbial activity. For instance, practices like reduced tillage, agroforestry, and rotational cropping can help maintain or increase soil organic carbon levels, which in turn stabilizes Rh. Additionally, incorporating organic farming techniques, such as the return of crop residues and the use of cover crops, can increase vegetation coverage and improve the carbon sequestration potential of the soil, thereby reducing Rh fluctuations. Furthermore, optimizing irrigation and fertilization practices can help maintain soil moisture and prevent nutrient imbalances that might otherwise lead to increased Rh. These measures could help buffer against extreme climate events and improve the resilience of ecosystems to changing environmental conditions, ultimately reducing the variability of Rh across regions. Therefore, rational planning of land use mode, restoration of degraded land and improvement of carbon sequestration capacity of regional ecosystems were important tasks in future land use planning. Regions with GDP higher per capita and population density had greater ability to realize rational land use planning.

## Supporting information

**S1 Data.**
(RAR)

## Author contributions

**Conceptualization:** Dan Liu.

**Data curation:** Dan Liu, Cheng Long Yu.

**Formal analysis:** Dan Liu.

**Funding acquisition:** Dan Liu.

**Methodology:** Rui Feng.

**Software:** Shi Ping Yin.

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
