## [Decision Letter · Decision Letter 0]

22 Jan 2025

PONE-D-24-54792Spatiotemporal dynamics and influencing factors of soil heterotrophic respiration in northeast ChinaPLOS ONE

Dear Dr. liu,

Thank you for submitting your manuscript to PLOS ONE. After careful consideration, we feel that it has merit but does not fully meet PLOS ONE’s publication criteria as it currently stands. Therefore, we invite you to submit a revised version of the manuscript that addresses the points raised during the review process.

We look forward to receiving your revised manuscript.

Kind regards,

Barathan Balaji Prasath

Academic Editor

PLOS ONE

“This research program was generously supported by the Natural Science Foundation of Heilongjiang Province (General Program) (Grant No.: LH2022D023), Innovation Development Project of China Meteorological Administration (Grant No.: CXFZ2023J059), and Key Laboratory Of Agrometeorological Disasters Joint Open Fund of Liaoning Provincial (Grant No.: 2023SYIAEKFZD06, 2023SYIAEKFMS27)”

“This research program was generously supported by the Natural Science Foundation of Heilongjiang Province (General Program) (Grant No.: LH2022D023), Innovation Development Project of China Meteorological Administration (Grant No.: CXFZ2023J059), and Key Laboratory of Agrometeorological Disasters Joint Open Fund of Liaoning Provincial (Grant No.: 2023SYIAEKFZD06, 2023SYIAEKFMS27)”

“This research program was generously supported by the Natural Science Foundation of Heilongjiang Province (General Program) (Grant No.: LH2022D023), Innovation Development Project of China Meteorological Administration (Grant No.: CXFZ2023J059), and Key Laboratory Of Agrometeorological Disasters Joint Open Fund of Liaoning Provincial (Grant No.: 2023SYIAEKFZD06, 2023SYIAEKFMS27)”

5. We note that Figures 4,5,6 and 7 in your submission contain [map/satellite] images which may be copyrighted. All PLOS content is published under the Creative Commons Attribution License (CC BY 4.0), which means that the manuscript, images, and Supporting Information files will be freely available online, and any third party is permitted to access, download, copy, distribute, and use these materials in any way, even commercially, with proper attribution. For these reasons, we cannot publish previously copyrighted maps or satellite images created using proprietary data, such as Google software (Google Maps, Street View, and Earth). For more information, see our copyright guidelines: http://journals.plos.org/plosone/s/licenses-and-copyright.

a. You may seek permission from the original copyright holder of Figures 4,5,6 and 7 to publish the content specifically under the CC BY 4.0 license. 

6. Please remove your figures from within your manuscript file, leaving only the individual TIFF/EPS image files, uploaded separately. These will be automatically included in the reviewers’ PDF.

Reviewers' comments:

Reviewer's Responses to Questions

**Comments to the Author**

1. Is the manuscript technically sound, and do the data support the conclusions?

Reviewer #1: Yes

Reviewer #2: Yes

2. Has the statistical analysis been performed appropriately and rigorously? 

Reviewer #1: No

Reviewer #2: Yes

3. Have the authors made all data underlying the findings in their manuscript fully available?

Reviewer #1: Yes

Reviewer #2: Yes

4. Is the manuscript presented in an intelligible fashion and written in standard English?

Reviewer #1: No

Reviewer #2: Yes

5. Review Comments to the Author

Reviewer #1: General Comments:

1. English Language: The manuscript requires a thorough check for grammar and language improvements.

2. Citation Pattern: Ensure the citation style is consistent throughout the manuscript.

Specific Comments:

1. Line 40: Correct citation format needed.

2. Line 54: Clarify whether this refers to laboratory calibration.

3. Lines 66 & 67: Avoid repetition; rewrite the sentence for clarity.

4. Line 71: Expand the term "Ra" when mentioned for the first time.

5. Line 73: Provide a reference as it is mentioned as "sensitive zone."

6. Materials and Methods Section: Include a map marking all relevant regions.

7. Lines 105 & 106: Rewrite the sentence for clarity and flow.

8. Line 155: Ensure citation adheres to journal instructions.

9. Lines 172 & 173: Clarify or rewrite the statement about soil respiration values.

10. Lines 186-190: Justify the inclusion of this section in "Materials and Methods" and mention any statistical analyses

performed.

11. Line 195: Correct spelling errors.

12. Lines 215 & 216: Justify the claim regarding minimal differences between grassland, farmland, and forest results with

statistical analysis.

13. Line 244: Rewrite the sentence for better clarity.

14. Lines 322, 329, 330, 331: Check for spelling errors.

15. Lines 364 & 365: Rewrite for better clarity.

16. Lines 383 & 384: Check for spelling errors and ensure proper citation as per journal instructions.

Reviewer #2: The manuscript entitled "Spatiotemporal dynamics and influencing factors of soil heterotrophic respiration in Northeast China" explores the spatiotemporal variations of soil heterotrophic respiration (Rh) in Northeast China from 2001 to 2020. It employs the GSMSR model, along with advanced spatial analysis techniques, to identify patterns and determine influencing factors, providing insights into regional carbon flux dynamics.

The manuscript is comprehensive, addressing a critical topic in soil science and carbon cycle research. It effectively integrates modeling, spatial analysis, and ecological insights. However, improvements in language clarity and the addition of more detailed methodological descriptions could enhance its academic quality.

General Comments:

1. Ensure consistency in units and terminology throughout the manuscript (e.g., kgC/ha/year vs kgC·ha⁻¹·yr⁻¹).

2. Revise the introduction to better align with the objectives and hypotheses.

3. Clarify the novelty of this research compared to similar studies using the GSMSR model.

4. Include more robust discussions on potential uncertainties in the model results.

5. Proofread the manuscript to address grammatical errors and enhance readability.

Line-to-Line Specific Comments:

1. Line 19: Revise "Soil heterotrophic respiration (Rh) is the main way of carbon output" to "Soil heterotrophic respiration (Rh) represents a primary pathway of carbon release from soil."

2. Line 27: Clarify the units used for carbon release ("4.76×10¹¹ kg/year") to ensure interpretability.

3. Line 30: Expand on the term "standard deviation ellipse" for readers unfamiliar with spatial statistics.

4. Line 62: Replace "calculate" with "calculated" for grammatical accuracy.

5. Line 73: Consider adding more context to explain why Northeast China is a "sensitive zone of global climate change."

6. Line 123: Ensure mathematical symbols in formulas are consistent with conventions (e.g., subscripts).

7. Line 148: Provide the reasoning for selecting 500m resolution for DEM data.

8. Line 163: Include a brief explanation of the GSMSR model's strengths compared to other models.

9. Line 171: Expand on the choice of static chamber/GC method for soil respiration measurement.

10. Line 196: Justify the use of quadratic curve fitting for calculating Rh.

11. Line 207: Explain why farmland has the highest Rh values among land use types.

12. Line 227: Clarify the significance of azimuth change range values in spatial analysis.

13. Line 271: Include ecological reasons for cold and hot spots observed in different regions.

14. Line 307: Justify excluding meteorological factors in the analysis of influencing factors.

15. Line 327: Provide references to support the claim that high vegetation coverage decreases soil aeration.

16. Line 339: Explain how local economic activities directly influence Rh rates with specific examples.

17. Line 350: Discuss potential errors introduced by assuming constant SOC in the model.

18. Line 370: Include a definition for NPP (Net Primary Productivity) for clarity.

19. Line 381: Expand on how land use optimization could mitigate Rh variability.

20. Line 403: Provide examples of human management modes affecting Rh.

21. Line 410: Suggest practical recommendations for improving vegetation coverage in degraded regions.

22. Figure 4 Legend: Improve the legend to ensure accessibility for readers unfamiliar with GIS methods.

23. Table 2: Highlight key findings or trends to guide readers in interpreting the data.

6. PLOS authors have the option to publish the peer review history of their article (what does this mean? ). If published, this will include your full peer review and any attached files.

**Do you want your identity to be public for this peer review?** For information about this choice, including consent withdrawal, please see our Privacy Policy .

Reviewer #1: No

Reviewer #2: **Yes: ** Fasih Ullah Haider

---

## [Author Response · Author response to Decision Letter 0]

10 Mar 2025

The responses to the reviewers' and editor's comments are saved in the 'Response to Reviewers.docx' file and have been uploaded earlier.

---

## [Decision Letter · Decision Letter 1]

31 Mar 2025

Spatiotemporal dynamics and influencing factors of soil heterotrophic respiration in northeast China

PONE-D-24-54792R1

Dear Dr.  Dan Liu,

We’re pleased to inform you that your manuscript has been judged scientifically suitable for publication and will be formally accepted for publication once it meets all outstanding technical requirements.

Kind regards,

Barathan Balaji Prasath

Academic Editor

PLOS ONE

Additional Editor Comments (optional):

Reviewers' comments:

Reviewer's Responses to Questions

**Comments to the Author**

1. If the authors have adequately addressed your comments raised in a previous round of review and you feel that this manuscript is now acceptable for publication, you may indicate that here to bypass the “Comments to the Author” section, enter your conflict of interest statement in the “Confidential to Editor” section, and submit your "Accept" recommendation.

Reviewer #1: All comments have been addressed

Reviewer #2: All comments have been addressed

2. Is the manuscript technically sound, and do the data support the conclusions?

Reviewer #1: Yes

Reviewer #2: Yes

3. Has the statistical analysis been performed appropriately and rigorously? 

Reviewer #1: Yes

Reviewer #2: Yes

4. Have the authors made all data underlying the findings in their manuscript fully available?

Reviewer #1: Yes

Reviewer #2: Yes

5. Is the manuscript presented in an intelligible fashion and written in standard English?

Reviewer #1: Yes

Reviewer #2: Yes

6. Review Comments to the Author

Reviewer #1: The corrections has been made.

The given comments were addressed

The references are corrected

The language is corrected

Reviewer #2: The authors have significantly improve the quality of paper according to comments and suggestions mentioned by reviewer, now the paper is suitable for publication.

7. PLOS authors have the option to publish the peer review history of their article (what does this mean? ). If published, this will include your full peer review and any attached files.

**Do you want your identity to be public for this peer review?** For information about this choice, including consent withdrawal, please see our Privacy Policy .

Reviewer #1: No

Reviewer #2: No

---

## [Editor Report · Acceptance letter]

PONE-D-24-54792R1

PLOS ONE

Dear Dr. Liu,

I'm pleased to inform you that your manuscript has been deemed suitable for publication in PLOS ONE. Congratulations! Your manuscript is now being handed over to our production team.

Kind regards,

on behalf of

Dr. Barathan Balaji Prasath

Academic Editor

PLOS ONE